# A Sampling Strategy to Develop a Primary Core Collection of *Miscanthus* spp. in China Based on Phenotypic Traits

Shuling Liu [1,†], Cheng Zheng [2,†], Wei Xiang [3,†], Zili Yi [1,4,*] and Liang Xiao [1,4,*]

1   College of Bioscience and Biotechnology, Hunan Agricultural University, Changsha 410128, China; yayaan@163.com
2   College of Agronomy, Hunan Agricultural University, Changsha 410128, China; chengzheng@stu.hunau.edu.cn
3   Crop Research Institute, Hunan Academy of Agricultural Science, Changsha 410128, China; xiangwei@hunaas.cn
4   Hunan Engineering Laboratory for Ecological Application of Miscanthus Resources, Hunan Agricultural University, Changsha 410128, China
*   Correspondence: yizili@hunau.net (Z.Y.); xiaoliang@hunau.edu.cn (L.X.)
†   These authors contributed equally to this work.

**Abstract:** Core collections can act as a genetic germplasm resource for biologists and breeders. Thirty-seven phenotypic traits from 471 Miscanthus accessions in China were used to design 203 sampling schemes to screen the genetic variations in different sampling strategies. The sampling was analyzed using the unweighted pair group method with arithmetic mean (*UPGMA*) and the Euclidean distance (*Euclid*). Several parameters including the variance of phenotypic value (*VPV*), Shannon–Weaver diversity index (*H*), coefficient of variation (*CV*), variance of phenotypic frequency (*VPF*), ratio of phenotype retained (*RPR*), the mean difference percentage (*MD%*) and the variance difference percentage of traits (*VD%*), the range coincidence rate (*CR%*) and the variable rate of quantitative traits (*VR%*) were used to evaluate the level of representation of the primary core collections developed by the different sampling schemes. Based on the optimal sampling strategies of prior selecting accessions, a primary core collection was constructed that maintained > 99.5% of the *VPV* and a *CR%* of 100%. This study indicates that the optimal sampling scheme consisted of prior and deviation sampling methods (PD) combined with a logarithmic proportional sampling strategy (LG) of 37.4% of the actual sampling ratio. Sampling before clustering can improve several parameters including the *H*, *CV*, *RPR*, *VPF*, and *CR%*. Sampling strategies including the genetic diversity index (G), logarithmic proportional (LG) and the square root proportional strategy (SG) can improve the *H*, whilst the constant strategy (C) can improve the *RPR* and *VPF* when the sampling scale was >30%. Furthermore, the proportional strategy (P) can improve the *VPV*.

**Keywords:** core collection; flora distribution; *Miscanthus* phenotypic trait

## 1. Introduction

Environmental concerns including the greenhouse effect and increasing demands for fossil fuels have stimulated research into renewable energy resources [1]. Indigenous materials have the potential to supply energy with lower emission of greenhouse gases and are more environmentally favorable [2]. One promising plant material that can be used to produce efficiently and economically biofuels with a lower land requirement is *Miscanthus* [3–5]. *Miscanthus* is a raw material candidate of lignocellulosic biomass [6] that is a perennial C₄ tall grass of the Gramineae Family, *Miscanthus* spp. It belongs to *subtribe Saccharinae Griseb., tribe Andropogoneae Dumort., Subfam. Panicoideae A. Braun* of *Poaceae* [7]. *Miscanthus* Andersson grows widely in Eastern and Southeastern Asia, the Pacific Islands, and Africa [8,9]. Fourteen different species of *Miscanthus* Andersson are found around the world of which seven different species are native to various provinces in China [8]. China

is the distribution center of the genus *Miscanthus*, and *M. lutorioriparius* Andersson is the endemic species of China [10].

*Miscanthus* is used in various industries including papermaking, animal feeds, and soil and water conservation [11]. Studies by the *Miscanthus* Research Institute of Hunan Agricultural University (HUNAU) have identified >3000 wild *Miscanthus* populations in >800 counties of 30 provinces in China since 2006. There are more than 1000 representative accessions for the seven native species that have been collected and grown in the *Miscanthus* germplasm garden in HUNAU. However, there is a need for a germplasm collection that can be used to improve the utilization and management of plant germplasm resources. Core collections can conserve germplasm collections and inform optimized plant breeding strategies. Whilst there have been extensive core collection efforts in species including wheat, rice, soybean, maize, sesame, and barley [12–18], there have been no reported studies on the core collection methods of Miscanthus.

Core collections can improve the conservation, evaluation and utilization of germplasm. Core collections selected as subsets can represent the maximum genetic diversity of the initial collection with the minimum redundancies [19]. The development of core collections includes the collection and analysis of data obtained from fields or greenhouses, implementing the principle of stratified sampling by dividing the accessions into different groups, determining the sampling proportion of each group within the core collection, the selection of samples at random or based on representative criteria and evaluating the diversity and representativeness of the core collection. Furthermore, for studies conducted using core collections, the most important procedure is the development of a robust sampling strategy including sampling scale, stratified principle, sampling proportion and the sampling method. In this study, we report on the sampling strategy of a *Miscanthus* primary core collection and its role in reducing the size of the initial collection whilst retaining genetic diversity in the collection.

## 2. Materials and Methods

### 2.1. Plant Materials

The research materials used in this study were a subset of Miscanthus collected by the *Miscanthus* Research Institute of HUNAU from different areas across China from 2006 to 2008. The collection consisted of >1000 accessions including *M. sinensis*, *M. floridulus*, *M. sacchariflorus*, and *M. lutarioriparius*. The materials were planted in red soil at the *Miscanthus* Germplasm Garden of HUNAU, Changsha, China (Lat. 28°11′ N, Long. 113°4′ E). Each accession was grown on a 2 m × 2 m plot. More than 60 quantitative traits relating to different developmental phases and various uses of the plant were measured annually. Four hundred and seventy-one accessions were used for sampling of the primary core collection that were originally located in 3 Kingdoms, 4 Sub Kingdoms, and 11 regions of China according to the Floristics of Seed Plants [20]. The numbers of accessions in each flora region are presented in Table S1.

### 2.2. Evaluation of Phenotypic Traits

Thirty-seven phenotypic traits were studied including 14 qualitative traits and 23 quantitative traits. The biological and agronomic trait data (Traits 1–25: Table 1) were collected during the reproductive stage. Yield and energy-related quality data (Traits 26–37: Table 1) were collected during the harvest season in December. The quantitative traits were used to calculate the mean ($X$) values and standard deviation ($\sigma$) to quantify the observation values ($X_i$) into categories. Each category represented one specific phenotype. The subdividing range of quantitative traits ranged from the category where $X_i > X - n \cdot \sigma$ to $X_i < X + m \cdot \sigma$ ($m > n$), with the interval between the two neighbor categories being 0.5 $\sigma$ (Table 2).

**Table 1.** The description and classes of the *Miscanthus* Phenotypic traits.

| | Traits | Abbreviation | Description and Classes |
|---|---|---|---|
| 1 | Date of bud emergence | DBE | Emergence date of second leaf |
| 2 | Date of beginning flowers | DBF | Flowering date of first flower |
| 3 | Days to beginning flowering | DsBF | Days from bud emergence to any plant produces flower |
| 4 | Plant height | PH | Height of largest over-ground complete plant |
| 5 | Stem length | SL | Length of over-ground complete stem |
| 6 | First internode length | FIL | First complete internode length of above-ground stem |
| 7 | stem axis long diameter of FIL | SALD | stem axis long diameter of FIL's middle |
| 8 | Node number of per stem | NS | Node number of per over-ground complete stem |
| 9 | Largest leaf length | LL | Length of visual largest leaf |
| 10 | Largest leaf width | LW | Width of visual largest leaves |
| 11 | Fresh weight of per stem | FWS | Fresh weight of stem after reproductive stage |
| 12 | Dry weight of per stem | DWS | Weighted after fresh stem was dried three days at 45 °C |
| 13 | Node hairiness | NH | Does node have hairiness? (No = "0", Yes = "1") |
| 14 | Leaf back hairiness | LBH | Does leaf back have hairiness? (No = "0", Yes = "1") |
| 15 | Sheath hairiness | ShH | Does sheath have hairiness? (No = "0", Yes = "1") |
| 16 | Sheath mouth hairiness | ShMH | Does sheath mouth have hairiness? (No = "0", Yes = "1") |
| 17 | Internode waxiness | IWa | Does internode have waxiness? (No = "0", Yes = "1") |
| 18 | Node waxiness | NWa | Does node have waxiness? (No = "0", Yes = "1") |
| 19 | Leaf waxiness | LWa | Does leaf have waxiness? (No = "0", Yes = "1") |
| 20 | Sheath waxiness | ShWa | Does sheath have waxiness? (No = "0", Yes = "1") |
| 21 | Stem color | StC | 0 = Yellow, 1 = Light green, 3 = Green, 5 = Dark green, 7 = lilac or pale-purple speckles interspersed; 9 = purple-red speckles interspersed |
| 22 | Leaf color | LC | 1 = Light green, 3 = Green, 5 = Dark green |
| 23 | Sheath color | ShC | 0 = Yellow, 1 = Light green, 3 = Green, 5 = Dark green, 7 = lilac or pale-purple speckles interspersed; 9 = purple-red speckles interspersed |
| 24 | Axillary bud on culm | ABC | 0 = No, 1 = Yes Does node have waxiness? |
| 25 | Angle of Stem | AS | 1 = Erect or $\theta \geq 80°$, 3 = $80° > \theta \geq 60°$, 5 = $60° > \theta \geq 40°$, 7 = $40° > \theta \geq 20°$, 9 = $\theta < 20°$ or Prostrate (Angle between plant outside stem and ground) |
| 26 | Tillers number per plot | TNP | Total number of tillers to plant on one plot |
| 27 | Dry matter content | DM | Dry matter content after fresh stem was dried to constant weight at 45 °C and at 105 °C |
| 28 | Neutral detergent fiber content | NDF | Determined with detergent fiber analysis |
| 29 | Acid detergent fiber content | ADF | Determined with detergent fiber analysis |
| 30 | Hemi-fibre content | HF | Determined with detergent fiber analysis |
| 31 | Fibre content | FC | Determined with detergent fiber analysis |
| 32 | Acid dissoluble lignin content | ADL | Determined with detergent fiber analysis |
| 33 | Acid insoluble ash content | AIA | Determined with detergent fiber analysis |
| 34 | Total ash content | TA | Ash content of matter incinerated in muffle furnace at 550 °C three hours |
| 35 | Total moisture content | TM | Total water content after fresh matter was dried to constant weight at 45 °C and at 105 °C |
| 36 | Total biomass per plot | TMP | Total biomass production of plants in one plot |
| 37 | Withered state | WiS | 0 = No, 1 = Yes (Have the plants begun to wither?) |

### 2.3. Sampling Strategy

The primary core collection was constructed using several methods based on grouping and ungrouping strategies (Figure 1). The ungroup-based strategy randomly selected three replicates from the initial collections. The primary core collection sampled using the random strategy was labeled as the non-group random sampling group (NGR). The group-based strategy involved a hierarchical two-level grouping approach in which each type of variety was grouped by flora after being grouped by species. In the hierarchical two-level grouping strategy, the accessions were divided into 23 hierarchical groups (Table S1).

Hierarchical two-level grouping methods and different sampling strategies were combined in this study. The primary core collections were selected from each group based on the given number of different sampling strategies. The clustering sampling methods were based on a stepwise clustering sampling method. A prior strategy of selecting accessions with the traits expressing maximum or minimum values as the primary core collections before clustering was used. The following sampling methods were used:

(1) A non-group random sampling method (NGR): In this method, the primary core collection was randomly selected from every subgroup with two germplasms at the lowest standard of categorizing. When one germplasm was in the subgroup, it was immediately selected for the cluster analysis. The procedures for the clustered and selected germplasms were repeated until the group scale was reduced to a given number.

(2) Deviation sampling (D): In this method, the degree of deviation degree of two germplasms were contrasted in each subgroup at the lowest standard of categorizing. The germplasm with the higher degree of deviation was selected for the following cluster analysis. When one germplasm was present in the subgroup, it was immediately selected for cluster analysis. The subsequent germplasms were processed similarly to the preceding step. The other procedures were similar to the stepwise clustering method.

**Table 2.** The number of Category for each trait.

| Quantitative Trait | Number of Category | Qualitative Trait | Number of Category |
|---|---|---|---|
| DBE | 7 | NH | 2 |
| DBF | 26 | LBH | 2 |
| DsBF | 29 | ShH | 2 |
| PH | 10 | ShMH | 2 |
| SL | 15 | IWa | 2 |
| FIL | 13 | NWa | 2 |
| SALD | 12 | LWa | 2 |
| NS | 14 | ShWa | 2 |
| LL | 10 | StC | 6 |
| LW | 13 | LC | 3 |
| FWS | 12 | ShC | 6 |
| DWS | 12 | ABC | 2 |
| TNP | 13 | AS | 5 |
| DM | 14 | WiS | 2 |
| NDF | 11 | - | - |
| ADF | 12 | - | - |
| HF | 14 | - | - |
| FC | 10 | - | - |
| ADL | 12 | - | - |
| AIA | 11 | - | - |
| TA | 14 | - | - |
| TM | 13 | - | - |
| TMP | 12 | - | - |

The degree of deviation of each quantitative trait was confirmed by the equation:

$$S_i^2 = \sum_{j=1}^{m} \frac{g_{ij}^2}{\sigma_j^2} \quad i = 1, 2, \ldots, n, \ j = 1, 2, \ldots, m \tag{1}$$

where $g_{ij}$ represents the $i$th value of the $j$th trait, and $\sigma_j^2$ represents the variance of the $j$th trait [21].

(3) Prior sampling (PR): Germplasms with the traits expressing the maximum or minimum values were chosen as core collections before clustering. The residual germplasms were processed using a method similar to the random clustering method.

(4) Prior and Deviation sampling (PD): This strategy was based on the prior sampling method. Germplasms were processed in a similar way to the deviation sampling method after the germplasms with the traits expressing the maximum or minimum values were selected as the core collections.

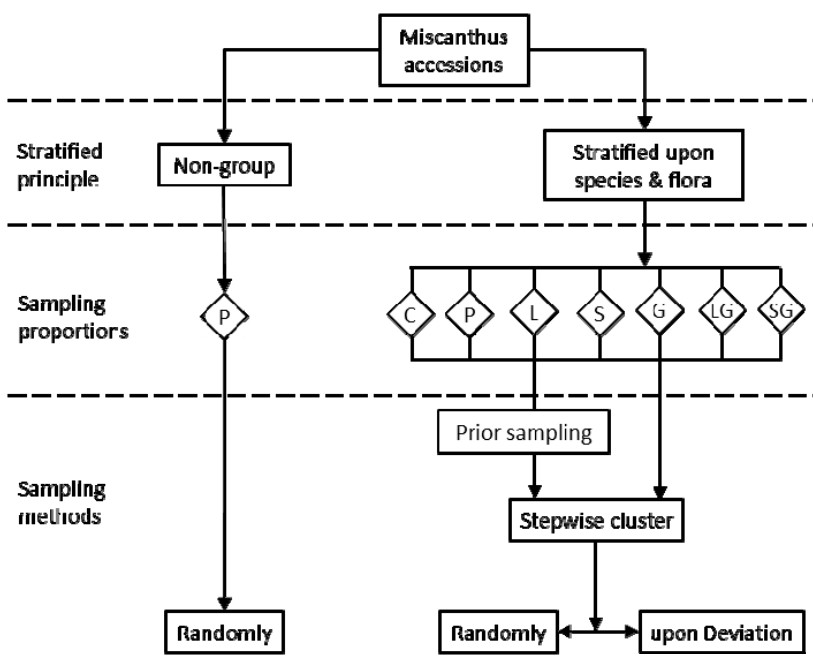

**Figure 1.** Sampling schemes of developing *Miscanthus* primary core collection. Constant strategy (C), Proportional strategy (P), Logarithm strategy (L), Square root strategy (S), Genetic diversity index strategy (G), Genetic diversity index adjusted with logarithmic proportional strategy (LG), Genetic diversity index adjusted with square root proportional strategy (SG). (1) Constant strategy (C)—the number of selected accessions sampled from each group was an equal number of accessions randomly; (2) Proportional strategy (P)—the number of selected accessions sampled from each group was proportional to the group size in the basic collection; (3) Logarithm strategy (L)—the number of selected accessions sampled from each group was proportional to the logarithmic group size in the basic collection; (4) Square root strategy (S)—sampling core collection from each group was proportional to the square root group size in the basic collection; (5) Genetic diversity index strategy (G)—sampling core collection from each group with proportional to genetic diversity index of the group in basic collection; (6) Genetic diversity index adjusted with logarithmic proportional strategy (LG)—sampling core collection from each group with the proportional to Shannon–Weaver diversity index was adjusted with logarithmic proportion; (7) Genetic diversity index adjusted with square root proportional strategy (SG)—sampling core collection from each group with the proportional to Shannon–Weaver diversity index was adjusted with square root proportion.

In the four clustering sampling methods, to reserve important biological types, it was decided that the groups including only one accession were selected as the primary core collection, e.g., two accessions of *M. floridulus* from the flora region of IIID12 and IVG22, one accession of *M. sinensis* from the flora region of IIIE14. Three accessions were selected as the primary core collections prior to clustering. Other groups were sampled using four clustering sampling methods. For comparison, the NGR was used to select a candidate primary core collection. Finally, 203 different sampling schemes were designed to develop a primary core collection of the *Miscanthus* in China.

To determine the optimal scale of a primary core collection, 20–50% ratios from the initial collections were considered as the ideal proportions for sampling. The actual numbers of selected accessions were calculated using different sampling proportions and combined with different sampling strategies and methods (Table 3).

*2.4. Evaluating the Parameters for the Core Collection*

Five parameters including *H*, *CV*, *VPV*, *VPF*, and the *RPR* were used to evaluate 203 sampling schemes [16].

**Table 3.** The sampling number of the primary core collections within different sampling strategies.

| Sampling Strategy | Ideal | | Actual | | | |
|---|---|---|---|---|---|---|
| | Number | Ratio (%) | Non-prior | Ratio (%) | Prior | Ratio (%) |
| C | 92 | 20 | 81 | 17.2 | 134 | 28.5 |
| | 115 | 25 | 98 | 20.8 | 141 | 29.9 |
| | 138 | 30 | 115 | 24.4 | 150 | 31.8 |
| | 161 | 35 | 131 | 27.8 | 159 | 33.8 |
| | 184 | 40 | 146 | 31.0 | 167 | 35.5 |
| | 207 | 45 | 161 | 34.2 | 177 | 37.6 |
| | 253 | 50 | 187 | 39.7 | 197 | 41.8 |
| G | 96 | 20 | 99 | 21.0 | 138 | 29.3 |
| | 119 | 25 | 120 | 25.5 | 150 | 31.8 |
| | 142 | 30 | 140 | 29.7 | 161 | 34.2 |
| | 165 | 35 | 160 | 34.0 | 174 | 36.9 |
| | 186 | 40 | 178 | 37.8 | 189 | 40.1 |
| | 212 | 45 | 198 | 42.0 | 203 | 43.1 |
| | 234 | 50 | 213 | 45.2 | 216 | 45.9 |
| L | 95 | 20 | 98 | 20.8 | 134 | 28.5 |
| | 116 | 25 | 119 | 25.3 | 145 | 30.8 |
| | 144 | 30 | 147 | 31.2 | 163 | 34.6 |
| | 163 | 35 | 166 | 35.2 | 177 | 37.6 |
| | 189 | 40 | 189 | 40.1 | 194 | 41.2 |
| | 211 | 45 | 210 | 44.6 | 210 | 44.6 |
| | 237 | 50 | 231 | 49.0 | 231 | 49.0 |
| LG | 93 | 20 | 96 | 20.4 | 130 | 27.6 |
| | 118 | 25 | 121 | 25.7 | 144 | 30.6 |
| | 142 | 30 | 145 | 30.8 | 159 | 33.8 |
| | 166 | 35 | 169 | 35.9 | 176 | 37.4 |
| | 188 | 40 | 191 | 40.6 | 194 | 41.2 |
| | 214 | 45 | 217 | 46.1 | 217 | 46.1 |
| | 235 | 50 | 237 | 50.3 | 237 | 50.3 |
| P | 95 | 20 | 98 | 20.8 | 127 | 27.0 |
| | 118 | 25 | 121 | 25.7 | 138 | 29.3 |
| | 142 | 30 | 146 | 31.0 | 157 | 33.3 |
| | 164 | 35 | 167 | 35.5 | 177 | 37.6 |
| | 187 | 40 | 190 | 40.3 | 198 | 42.0 |
| | 211 | 45 | 214 | 45.4 | 220 | 46.7 |
| | 244 | 50 | 244 | 51.8 | 248 | 52.7 |
| S | 97 | 20 | 97 | 20.6 | 131 | 27.8 |
| | 116 | 25 | 116 | 24.6 | 140 | 29.7 |
| | 143 | 30 | 140 | 29.7 | 155 | 32.9 |
| | 165 | 35 | 162 | 34.4 | 170 | 36.1 |
| | 186 | 40 | 181 | 38.4 | 185 | 39.3 |
| | 212 | 45 | 206 | 43.7 | 208 | 44.2 |
| | 236 | 50 | 227 | 48.2 | 228 | 48.4 |
| SG | 95 | 20 | 98 | 20.8 | 129 | 27.4 |
| | 120 | 25 | 123 | 26.1 | 142 | 30.1 |
| | 143 | 30 | 146 | 31.0 | 157 | 33.3 |
| | 164 | 35 | 167 | 35.5 | 174 | 36.9 |
| | 188 | 40 | 191 | 40.6 | 194 | 41.2 |
| | 213 | 45 | 216 | 45.9 | 218 | 46.3 |
| | 234 | 50 | 237 | 50.3 | 238 | 50.5 |

Note: Constant strategy (C), Proportional strategy (P), Logarithm strategy (L), Square root strategy (S), Genetic diversity index strategy (G), Genetic diversity index adjusted with logarithmic. Proportional strategy (LG), Genetic diversity index adjusted with square root proportional strategy (SG).

The mean percentage difference (*MD*%), variance percentage difference (*VD*%), range coincidence rate (*CR*%) and variable rate (*VR*%) of the quantitative traits were compared by assessing the optimal sampling strategy [22]:

$$CR\% = \frac{1}{m} \sum_{j=1}^{m} \frac{R_C}{R_I} \times 100$$

$$VR\% = \frac{1}{m} \sum_{j=1}^{m} \frac{CV_C}{CV_I} \times 100$$

(2)

where $M_C$ = the mean of the core collection, $M_I$ = the mean of the initial collection, $R_C$ = the average scope of the quantitative traits of core collections, $R_I$ = the average range of the quantitative traits of the initial collections, $CV_C$ = the coefficient of variation of traits for the core collections, $CV_I$ = the coefficient of variation of traits for the initial collection, $m$ = the number of the quantitative traits.

Core collections are required to meet two criteria to accurately represent the genetic diversity of the initial collection. Specifically, core collections should include ≤20% of the traits possessed by diverse means ($\alpha$ = 0.05) between the core and initial collections, and the core collection should retain a range coincidence rate (*CR*%) ≥80% of the traits [23,24].

In developing the primary core collection, the four sampling methods (i.e., R, D, PR, and PD), seven sampling strategies (i.e., C, P, L, S, G, LG, and SG), and seven different sampling proportions (20%, 25%, 30%, 35%, 40%, 45%, and 50%) were applied. Then, 203 potential primary core collections were constructed and denoted as R-C20, D-P25, PR-L30, PD-LG35, etc. In contrast, seven non-group primary core collections were constructed using a combined proportional strategy (P) with different sampling proportions that were denoted as NGR20 to NGR50.

## 3. Results

### 3.1. The Tendency of Parameters for Sampling Methods in Different Sampling Scales

The variation tendency of the five parameters obtained using five sampling methods at seven sampling scales was processed (Figure 2). The group-based strategy was shown to be superior to the non-group strategy, and the methods of sampling before the clustering methods (PD and PR) were superior to the other clustering methods. The prior sampling strategy was potentially optimal for sampling. The variances of phenotypic value (*VPV*) of the primary core collections increased when reducing the sampling scale. The primary core collections constructed by the group strategy had similar *VPV*. Furthermore, the *VPVs* were all higher compared to the primary core collections constructed by NGR (Figure 2a). The *H* of the PD and PR methods increased with reducing sampling scale, yet the *H* of the method of deviation sampling (D) and random clustering (R) decreased with reducing sampling scale. The *VPV* of the NGR method showed no obvious regularity (Figure 2b). The coefficient of variation (*CV*) of the primary core collections constructed using various sampling methods showed undulating changes at a high sampling scale that then declined at a lower scale (Figure 3c). The ratio of phenotype retained (*RPR*) of the PD and PR methods was similar to the different sampling scales. The *RPR* of other methods decreased with reducing sampling scale (Figure 2d). The tendency of the ratio of variance of the phenotypic frequency (*VPF*) increased with a reducing sampling scale (Figure 2e). The *H*, *CV* and *RPR* of the PD and PR methods were similar or higher than the parameters of the other methods. The *H*, *CV* and *RPR* of the D and R methods were similar and higher than the NGR method. The *VPFs* of PD and PR methods were almost the same and lower than methods D and R. The group strategy was superior to the NGR, and the *VPV* of the prior strategy was superior to those of other strategies. In conclusion, the clustering methods of the P and D sampling methods were optimal.

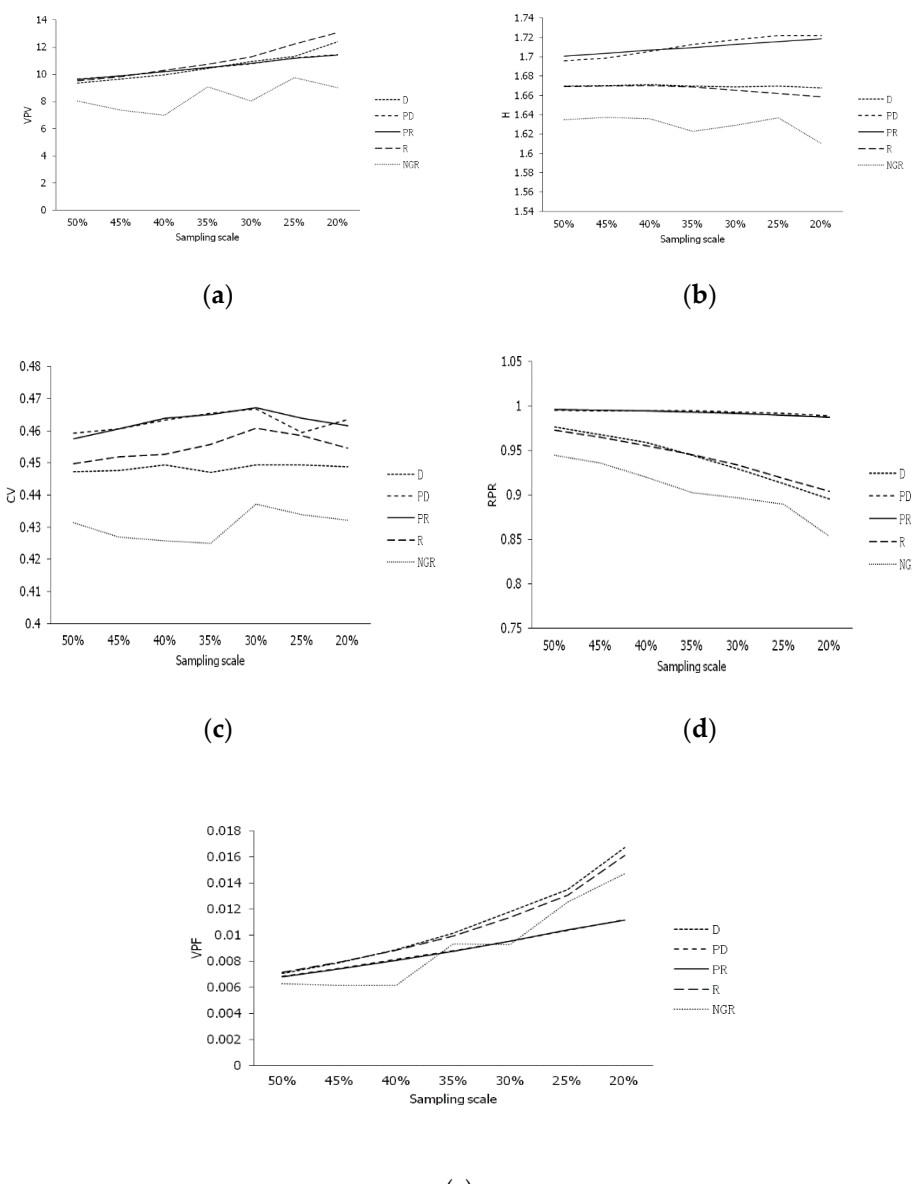

**Figure 2.** Tendency of parameters for sampling methods in different sampling scales. (**a**) Tendency of *VPV*; (**b**) Tendency of *H*; (**c**) Tendency of *CV*; (**d**) Tendency of *RPR*; (**e**) Tendency of *VPF*. *H*: Shannon–Weaver diversity index, *CV*: coefficient of variation, *VPV*: variance of phenotypic value, *VPF*: variance of phenotypic frequency, *RPR*: ratio of phenotype retained. D, PD, PR, R, and NGR stand for deviation sampling (D), prior and deviation sampling (PD), prior sampling (PR), random clustering (R), and non-group random sampling method (NGR), respectively.

*3.2. The Tendency of the Parameters for Sampling Strategies in Different Sampling Scales*

The five parameters obtained from the seven sampling strategies at the seven sampling scales were compared (Figure 3). The *VPV* for various core collections increased with a reduced sampling scale. The *VPV* of the core collections was highest when constructed using the constant strategy (C) and lowest when using the proportional strategy (P) (Figure 3a). The general tendency of *H* increased with reduced sampling scales. The value of *H* fluctuated when the sampling scale was <30% (Figure 3b). The tendency of the *CV* had no obvious regularity and mostly increased at a high sampling scale and decreased at a lower scale (Figure 3c). The *RPR* of all methods was similar and decreased with reducing sampling scales (Figure 3d). The *VPF* increased with reduced sampling scales and the C strategy was inferior to other strategies (Figure 3e). The *RPR* and *VPF* of the C strategy

and the *VPV* and *H* of the P strategy performed the worst. These data indicated that the two sampling strategies were not applicable. Sampling strategies G, LG, SG, L, and S could potentially be used.

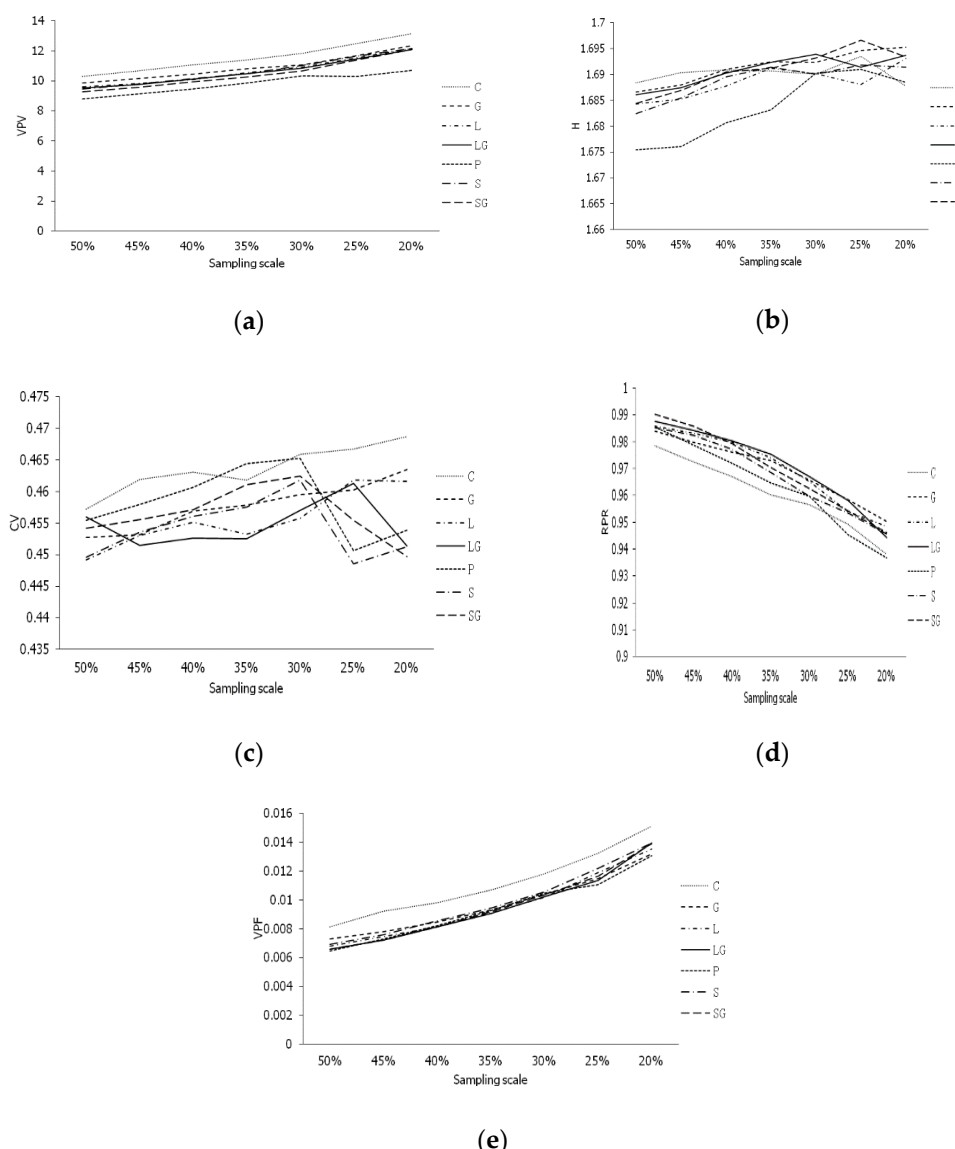

**Figure 3.** Tendency of parameters for sampling strategies in different sampling scales. (**a**) Tendency of *VPV*; (**b**) Tendency of *H*; (**c**) Tendency of *CV*; (**d**) Tendency of *RPR*; (**e**) Tendency of *VPF*. Shannon–Weaver diversity index (*H*), coefficient of variation (*CV*), variance of phenotypic value (*VPV*), variance of phenotypic frequency (*VPF*), and ratio of phenotype retained (*RPR*). C, G, L, LG, P, S, and SG stand for Constant strategy (C), Genetic diversity index strategy (G), Logarithm strategy (L), Genetic diversity index adjusted with logarithmic proportional strategy (LG), Proportional strategy (P), Square root strategy (S), Genetic diversity index adjusted with square root proportional strategy (SG), respectively.

### 3.3. The Relationship of the Parameters between Different Sampling Strategies and Methods

The five parameters obtained from the four clustering methods used in the different sampling strategies were compared at different sampling scales (Figure 4). From the results, the prior sampling strategy methods led to improved effectiveness in *H*, *CV*, *RPR*, and *VPF* amongst the different sampling strategies. The *VPV* values calculated for the four sampling methods were similar (Figure 4a). The *H*, *CV*, and *RPR* calculated from the primary core collections using the prior sampling strategy were higher than those for the other

sampling strategies. The *VPF* calculated from the core collections using the prior sampling strategy was lower than for the other sampling strategies (Figure 4b–e). There were no significant differences in the five parameters between the PD and PR clustering methods. Prior sampling before clustering resulted in higher *H*, *CV* and *RPR* but a lower *VPF* of the primary core collections compared to the other two sampling methods. These data indicate that the methods of prior sampling before clustering were superior to directly clustering.

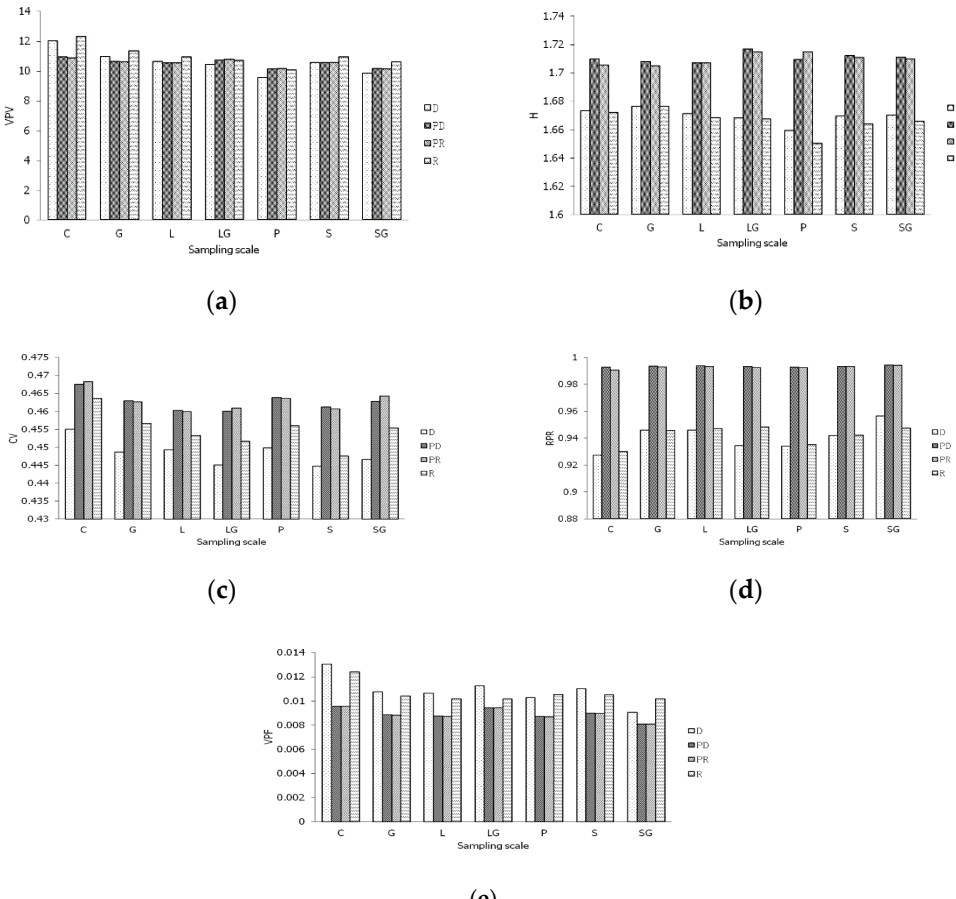

**Figure 4.** Difference of parameters for sampling strategies in different sampling scale. (**a**) Difference of *VPV*; (**b**) Difference of *H*; (**c**) Difference of *CV*; (**d**) Difference of *RPR*; (**e**) Difference of *VPF*. Shannon–Weaver diversity index (*H*), coefficient of variation (*CV*), variance of phenotypic value (*VPV*), variance of phenotypic frequency (*VPF*), and ratio of phenotype retained (*RPR*). D, PD, PR, R stand for deviation sampling (D), prior and deviation sampling (PD), prior sampling (PR), random clustering (R), respectively.

### 3.4. Comparison of the Sampling Strategies and Methods

The five parameters of the different sampling proportions, strategies and methods within the group were compared using Duncan's multiple range tests. The results are presented in Table 4 where the same ranking score implies that the data were not significantly different. The different ranking scores indicate superior to inferior assets.

Seven types of sampling strategies were compared across the groups using hierarchical cluster sampling. The results indicated that sampling according to the genetic diversity index strategy (G) was optimal, followed by the genetic diversity index adjusted with logarithmic proportional strategy (LG) and the genetic diversity index adjusted with a square root proportional strategy (SG). The square root strategy (S) gave the worst results. The ranking of the sampling schemes strategies from superior to inferior was G > LG > SG > C > L > P > S. In the same table, five sampling methods were compared. The results indicated that the hierarchical cluster methods were superior to the NGR methods.

The non-group-based strategy was performed worst. The prior sampling strategies, PD and PR, performed better than the non-prior sampling strategies (D, R and NGR). The superior-to-inferior order of the sampling schemes was PD > PR > D > R > NGR.

**Table 4.** The rank of sampling strategies, sampling methods and sampling scales within group in 203 primary core collections.

| Parameter | Sampling Strategy | | | | | | | Sampling Method | | | | |
|---|---|---|---|---|---|---|---|---|---|---|---|---|
| | C | G | L | LG | P | S | SG | PR | PD | R | D | NGR |
| *VPV* | 1 | 2 | 4 | 5 | 7 | 3 | 6 | 4 | 3 | 2 | 1 | 5 |
| *H* | 4 | 1 | 6 | 3 | 7 | 5 | 2 | 2 | 1 | 3 | 4 | 5 |
| *VPF* | 1 | 3 | 5 | 6 | 2 | 7 | 4 | 1 | 2 | 4 | 3 | 5 |
| *RPR* | 7 | 5 | 4 | 2 | 1 | 6 | 3 | 1 | 2 | 5 | 4 | 3 |
| *CV* | 7 | 4 | 2 | 1 | 6 | 5 | 3 | 2 | 1 | 4 | 3 | 5 |
| Sum of rank | 20 | 15 | 21 | 17 | 23 | 26 | 18 | 10 | 9 | 18 | 15 | 23 |

Note: Shannon–Weaver diversity Index (*H*), coefficient of variation (*CV*), variance of phenotypic value (*VPV*), variance of phenotypic frequency (*VPF*), and ratio of phenotype retained (*RPR*), Constant strategy (C), Proportional strategy (P), Logarithm strategy (L), Square root strategy (S), Genetic diversity index strategy (G), Genetic diversity index adjusted with logarithmic proportional strategy (LG), Genetic diversity index adjusted with square root proportional strategy (SG). PR, PD, R, D, NGR stand for prior sampling, prior and deviation sampling, random clustering, deviation sampling, and non-group random sampling method (NGR), respectively.

The averages of the ranking scores of the five parameters of all 203 sampling schemes combining the different sampling strategies with different sampling methods are summarized in Table 5. When comparing the 203 sampling schemes based on sampling strategies, we found that the L and LG sampling methods of PD had the highest scores. The sampling scheme of PD-LG resulted in the highest score among all schemes.

**Table 5.** Comparison of the average ranking scores of the five parameters for sampling strategies with methods.

| Sampling Strategies | Sampling Methods | | | | | Average |
|---|---|---|---|---|---|---|
| | D | R | PD | PR | NGR | |
| C | 19.0 | 16.2 | 8.8 | 10.2 | - | 13.55 |
| G | 17.6 | 15.6 | 8.4 | 11.2 | - | 13.20 |
| L | 19.6 | 16.2 | 8.2 | 8.2 | - | 13.05 |
| LG | 17.6 | 17.4 | 7.8 | 9.4 | - | 13.05 |
| P | 25.0 | 25.6 | 11.8 | 10.0 | 25.8 | 19.64 |
| S | 22.8 | 19.0 | 9.2 | 10.8 | - | 15.45 |
| SG | 23.6 | 18.6 | 10.2 | 10.6 | - | 15.75 |
| Average | 20.74 | 18.37 | 9.20 | 10.06 | 25.80 | 14.81 |

Note: Random clustering (R), Deviation sampling (D), Prior sampling (PR), Prior and Deviation sampling (PD), non-group random sampling method (NGR). Constant strategy (C), Proportional strategy (P), Logarithm strategy (L), Square root strategy (S), Genetic diversity index strategy (G), Genetic diversity index adjusted with logarithmic proportional strategy (LG), Genetic diversity index adjusted with square root proportional strategy (SG).

### 3.5. Comparison of the Sampling Scale of the Core Collection

Comparison between the seven sampling scales showed that scales of 25%, 30% and 35% performed significantly better than the other sampling scales and followed the order of 30% > 25% = 35% > 40% > 20% = 45% > 50% (Table 6). The *CR*% increased with increasing sampling scale except for the *CR*% from 20% and 25% sampling proportions combined with sampling methods (Table 7). Furthermore, the sampling strategies did not influence the *CR*% results. The *CR*% values reached 100% when the sampling scales were >35%.

### 3.6. Assessment of the Core Collections with 21 Quantitative Traits

The results from different sampling schemes are summarized in Table S2. Of these, 176 primary core collections had 100% *VD*%. The *MD*% of these accessions was significantly different (*MD*% $\geq$ 33.3%) from the initial collections. All the *CR*% values were >80% and 96 of those reached 100% indicating a high range of variation of the traits. Prior sampling before clustering gave the largest *CR*% values. The *VD*% of deviation sampling strategies combined with prior sampling were lower than the random sampling strategies. The

*VR*% of the grouped sampling core collections were >100% and 53 *VR*% of the primary core collections had >110%. These data may be caused by the increased variation of traits after removing redundant germplasms by sampling germplasms with the traits expressing maximum or minimum values prior. Twenty (PD-LG35, PD-S35, PR-LG30, PD-P30, PD-SG30, PR-P30, PR-SG25, PD-P25, PR-P25, PD-L20, PR-C20, PR-L20, PD-S20, PR-S20, PD-LG20, PR-LG20, PD-SG20, PR-SG20, PD-P20, PR-P20) core collections had the highest *VD*% and *CR*% values, the lowest *MD*%, and the higher *VR*% in which PD-LG35 had the largest number of accessions.

**Table 6.** Sum of the rank of sampling scales within groups in 203 primary core collections.

| Parameter | Sampling Scale | | | | | | |
|---|---|---|---|---|---|---|---|
| | **20%** | **25%** | **30%** | **35%** | **40%** | **45%** | **50%** |
| *VPV* | 1 | 2 | 3 | 4 | 5 | 6 | 7 |
| *H* | 2 | 1 | 3 | 4 | 5 | 6 | 7 |
| *VPF* | 5 | 3 | 1 | 2 | 4 | 6 | 7 |
| *RPR* | 7 | 6 | 5 | 4 | 3 | 2 | 1 |
| *CV* | 7 | 6 | 5 | 4 | 3 | 2 | 1 |
| Sum of rank | 22 | 18 | 17 | 18 | 20 | 22 | 23 |

Note: Shannon–Weaver diversity index (*H*), coefficient of variation (*CV*), variance of phenotypic value (*VPV*), variance of phenotypic frequency (*VPF*), and ratio of phenotype retained (*RPR*).

**Table 7.** Comparison of the range coincidence rates (CR%) of the primary core collections.

| | | Sampling Scale | | | | | | |
|---|---|---|---|---|---|---|---|---|
| | | **20%** | **25%** | **30%** | **35%** | **40%** | **45%** | **50%** |
| Sampling Method | D | 90.24 | 96.40 | 91.18 | 96.28 | 100.00 | 100.00 | 100.00 |
| | R | 92.35 | 97.53 | 92.46 | 97.32 | 100.00 | 100.00 | 100.00 |
| | PD | 94.38 | 98.01 | 93.85 | 100.00 | 100.00 | 100.00 | 100.00 |
| | PR | 95.67 | 89.76 | 95.30 | 100.00 | 100.00 | 100.00 | 100.00 |
| Sampling Strategy | C | 94.09 | 95.14 | 96.18 | 96.37 | 97.22 | 98.13 | 98.40 |
| | G | 95.45 | 96.28 | 96.69 | 98.13 | 98.28 | 98.40 | 98.95 |
| | LG | 94.91 | 95.98 | 96.35 | 97.81 | 97.96 | 98.76 | 98.95 |
| | SG | 95.54 | 96.29 | 96.87 | 97.10 | 98.04 | 98.91 | 99.03 |
| | L | 95.18 | 95.97 | 97.32 | 98.15 | 98.39 | 98.82 | 99.01 |
| | P | 94.91 | 95.56 | 96.34 | 96.74 | 97.25 | 97.48 | 98.60 |
| | S | 94.91 | 95.98 | 97.21 | 97.36 | 98.32 | 98.68 | 98.89 |

Note: Random clustering (R), Deviation sampling (D), Prior sampling (PR), Prior and deviation sampling (PD), Constant strategy (C), Proportional strategy (P), Logarithm strategy (L), Square root strategy (S), Genetic diversity index strategy (G), Genetic diversity index adjusted with logarithmic proportional strategy (LG), Genetic diversity index adjusted with square root proportional strategy (SG).

*3.7. Determination of the Sampling Scheme of the Core Collection*

The *H*, *CV* and *RPR* of the primary core collections developed according to the combined PD and PR and G and LG strategies within all the sampling proportions are compared in Table 8. From Table S3, the *RPR* of all the candidate core collections were reduced by reducing the proportion of sampling, whilst the *H* and *CV* increased by reducing the sampling proportion. The *RPRs* were about 98.8%, 99.2%, 99.4%, 99.5% and 99.6%, respectively, two of which have reached 99.6%. No significant difference between those of all candidate primary core collections. The *H* and *CV* were larger compared to the initial collections.

The results of the sampling schemes were grouped using hierarchical clustering methods of the PD and PR and G and LG sampling strategies as summarized in Table S3. The rank of *VPV*, *H*, *CV*, *VPF* and *RPR* of all the sampling ratios from the whole collections indicated that the PD sampling method combined with the LG sampling strategy performed best at a sampling proportion of 35%. This sampling scheme developed a core collection with 176 accessions in which the actual sampling ratio is 37.4% (Table S3).

**Table 8.** Comparison of the sampling rations in candidate primary core collections.

| Parameter | Sampling Scheme | Sampling Scale | | | | | | |
|---|---|---|---|---|---|---|---|---|
| | | **20%** | **25%** | **30%** | **35%** | **40%** | **45%** | **50%** |
| *H* | PD-G | 1.719 | 1.716 | 1.710 | 1.708 | 1.705 | 1.700 | 1.698 |
| | PD-LG | 1.724 | 1.725 | 1.721 | 1.716 | 1.706 | 1.698 | 1.699 |
| | PR-G | 1.713 | 1.709 | 1.702 | 1.704 | 1.702 | 1.704 | 1.701 |
| | PR-LG | 1.722 | 1.717 | 1.714 | 1.713 | 1.709 | 1.705 | 1.702 |
| *CV* | PD-G | 46.764 | 46.513 | 46.566 | 46.267 | 46.186 | 45.938 | 45.802 |
| | PD-LG | 46.717 | 46.409 | 46.087 | 46.066 | 45.961 | 45.745 | 45.193 |
| | PR-G | 46.726 | 46.637 | 46.793 | 46.122 | 46.179 | 45.869 | 45.511 |
| | PR-LG | 46.932 | 46.359 | 46.369 | 45.858 | 45.775 | 45.489 | 45.147 |
| *RPR* | PD-G | 98.900 | 99.300 | 99.400 | 99.400 | 99.400 | 99.500 | 99.500 |
| | PD-LG | 98.900 | 99.300 | 99.400 | 99.500 | 99.500 | 99.500 | 99.500 |
| | PR-G | 98.800 | 98.800 | 99.300 | 99.400 | 99.400 | 99.500 | 99.600 |
| | PR-LG | 98.700 | 99.100 | 99.300 | 99.500 | 99.500 | 99.500 | 99.600 |

Note: Shannon–Weaver diversity index (*H*), coefficient of variation (*CV*), and ratio of phenotype retained (*RPR*). PD-G, PD-LG, PR-G, PR-LG stand for prior and deviation sampling method combined with genetic diversity index strategy, prior and deviation sampling method combined with logarithmic proportional strategy, prior sampling method combined with genetic diversity index strategy, prior sampling method combined with logarithmic proportional strategy, respectively.

## 4. Discussion

### 4.1. Phenotype Data Construction of a Primary Core Collection

The aim of developing a core collection is to build a population with minimal samples whilst maintaining maximum genetic diversity. Many core collections of crop germplasms have been successfully constructed including rice, wheat, soybean, and other commercial crops [14–30]. Currently, several types of data are used to construct core germplasm collections including habitat, phenotypic, and genomic data [31]. The distribution information and biological and agronomic traits were used in this study. It is difficult to establish core collections by assessing the genetic diversity of a whole germplasm resource using phenotype traits. Although molecular markers have been used for evaluating genetic diversity at the DNA level in crop germplasm resources [32], the application of such approaches to entire collections is laborious and costly. The development of primary core collections based on phenotype traits could reduce the scale of entire collections along with labor intensity and costs.

Phenotypic data has been previously used to build core collections in Miscanthus [33]. This approach showed that the grouping method based on the original geography data was the best strategy compared with the other grouping methods such as single phenotypic, random, administrative province, and non-grouping methods. In this study, we used phenotype data to establish core collections using different strategies in Miscanthus. We used five parameters including *H*, *CV*, *VPV*, *VPF*, and *RPR* to screen 203 candidate core collections. Our results showed that the PD-LG35 sampling strategy (prior and deviation sampling method, genetic diversity index adjusted with logarithmic proportional strategy, and 35% sampling ratio) was used to develop a core collection with 176 accessions, had the highest genetic diversity and optimum number of samples. Considering the data collected from the same observation station named Miscanthus germplasm garden built in 2006 in Hunan agricultural university [34], theoretically believe that all germplasm growth was in the same environment, therefore, the difference of phenotype traits able to stand for the genetic variation among individuals.

### 4.2. The Method to Establish the Primary Core Collection

Sampling strategies are a key factor in establishing a primary core collection. Studies have used different approaches to construct primary core collections such as the proportional strategy (P) in the apricot germplasm in China [28] and the genetic diversity index

strategy (G) in safflower germplasm [35]. The scale of the sampling ratio is also an important factor that impacts the efficiency of primary core collections. Moreover, the scale of the primary core collection to the whole collection should be determined according to the size of the initial collection group. The sampling proportions may vary depending on the size of the initial collections. In spite of previous studies not suggesting any referable ration or any appropriate size for the primary core collection of *Miscanthus*, the ratio of the core collection to the whole collection for core collections established worldwide for different species is around 5–30% [26,36,37]. According to our preliminary study of the sampling strategy of *Miscanthus* in China, core collections of sampling before selecting core collections strategies retain a higher proportion of the phenotype characteristics (*RPR* > 98.8%). The PD-LG at 35% sampling proportion had the highest *H* and *CV* in the schemes compared to the PD-LD at 40%, 45%, and 50% which had the same or larger *RPR*.

Our data show that the group-based strategy was superior to the non-group strategy in different sampling scales or sampling strategies. Germplasm materials with similar heredity characteristics can be classified as one group using the group-based strategy. The methods of prior sampling before clustering methods were superior to the other clustering methods in different sampling scales because the germplasms with greater research value and special traits were not excluded. The *VPV* calculated based on the four sampling methods on different sampling scales or using sampling strategies were very similar and may be attributed to the rich genetic diversity of *Miscanthus* caused by intraspecific crossing. The constant strategy performed the worst at different sampling scales and may be attributed to the nonuniform genetic diversity of the intra-group *Miscanthus* as well as the proportional strategy (P). The sampling according to G, LG, and SG gave better results probably due to the affirmation of sampling ratio according to genetic diversity.

## 5. Conclusions

The PD-LG35 sampling strategy was used to develop a primary core collection with 176 accessions that had the best performance in this study. The actual sampling ratio was 37.4% suggesting that this was the optimal sampling scheme for selecting core collections. With such a moderate number of *Miscanthus* in China, PD methods combined with the LG at 37.4% of the actual sampling ratio was the optimum strategy. Furthermore, prior sampling before clustering could improve *H*, *CV*, *RPR* and *VPF*, with little impact on *VPV*. This sampling strategy also could improve the range of the *CR*% without affecting on the *MD*%. The sampling strategies using G, LG, and SG could improve *H*. Meanwhile, the C had the disadvantage of improving the *RPR* and *VPF* when the sampling scale was more than 30%, whilst the P had the disadvantage of improving the *VPV*.

**Supplementary Materials:** The following supporting information can be downloaded at: https://www.mdpi.com/article/10.3390/agronomy12030678/s1, Table S1: The number of accessions in each flora and species; Table S2: Comparison of the percentages for the differences between the primary core collections and the initial collections; Table S3: Rank of the integrative score of 5 parameters from candidate core collections. Shannon–Weaver diversity index (*H*), coefficient of variation (*CV*), variance of phenotypic value (*VPV*), variance of phenotypic frequency (*VPF*), and ratio of phenotype retained *(RPR)*.

**Author Contributions:** Conceptualization, Z.Y.; methodology, S.L.; software, S.L. and C.Z.; validation, W.X. and C.Z.; investigation, L.X.; writing—original draft preparation, S.L. and W.X.; writing—review and editing, L.X.; funding acquisition, Z.Y. All authors have read and agreed to the published version of the manuscript.

**Funding:** This study was financially supported by the Foundation for the Construction of the Innovative Hunan Province (grant number: 2019NK2011).

**Institutional Review Board Statement:** Not applicable.

**Informed Consent Statement:** Not applicable.

**Data Availability Statement:** The data used to support the findings of this study are available from the corresponding author upon request.

**Acknowledgments:** We thanks to these M.D candidate students during 2009 and 2012 in Hunan Agricultural University, who collected data in the field, included Cong lin, De Xue, Bin hu, Zuhong Wang, Yueyue Zhou, Yu Wang, and Guote Deng.

**Conflicts of Interest:** The authors declare no conflict of interest.

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
