# Peer review of "A Sampling Strategy to Develop a Primary Core Collection of Miscanthus spp. in China Based on Phenotypic Traits"

_agronomy, doi:10.3390/agronomy12030678_

Round 1

Reviewer 1 Report

The literature review is to be corrected. I do not recommend quoting [9-13] because it does not make sense. Please replace the items quoted from the last century with contemporary ones. Why in the methodology in Table 1, some of the results have +/- SD and some of them do not have such deviations from the mean values? The values in parentheses should be set to which place after the decimal point and write them in this way. Table 2 and 3 without statistical errors? What was the method of assessing these parameters? Results table 4 - some statistical study? All tables - SD should not be written in parentheses only as: mean value +/- SD + statistic (post hoc test) Please work on your conclusions

Reviewer 2 Report

The manuscript “Sampling Strategy to Develop a Primary Core Collection of Miscanthus spp. in China Based on Phenotypic Traits” is an interesting report regarding strategies for developing Miscanthus core collections. The efficient use of genetic resources in genetic breeding is crucial for developing elite germplasm. Establishing a core collection can help increase the efficient use of this germplasm. Hence, the manuscript could potentially be accepted for publication if the authors meet the major suggestions stated below:

Major suggestions (it may affect the structure of the manuscript):

I am not completely qualified to judge the English. However, the manuscript should be submitted to a careful grammatical and spelling review. I believe the authors made a good effort to minimize them, but I suggest the authors do another round of English reviews. This is critically important before the acceptance of the manuscript. As it is right now, the manuscript is not suitable for publishing in Agronomy – a journal with an international audience.

The manuscript seems to be a pre-submission version. There are many sections in the text that still contains “tracked-changes”. I could not find any “clean” version for review. Therefore, I assume this was the submitted version. These “tracked-changes” only can be used alongside a “clean” version. Otherwise, the “tracked-changes” can impair the readability of the text and, consequently, the reviewing process.

In the Material and Methods and Results, the excessive use of abbreviations for designating the sampling strategies and the statistics make the reading complex. The use of the ampersand (&) makes the text even more complex. As an example, consider the following sentence: “The RPR & VPF of C strategy and the VPV & H of P strategy performed the worst respectively, indicating that the two sampling strategies were not applicable, and sampling strategies of G, LG, SG, L and S were potential strategies”. Although the abbreviations were correctly declared, the excess of abbreviations can make the text hard to read. Abbreviations should be used, indeed. However, in excess, they unnecessarily increase the complexity of the text.

Minor suggestions:

L3 – The word “Miscanthus” used to refer to the genus Miscanthus should be formatted in italic or bold.

L77 – “The research materials” -> “The research material”

L96 – “to quantified” -> “to quantify”

L95:L97 – This sentence is not clear. I suggest the authors to rephrase this sentence, since it is not being possible to understand what “values of quantitative traits” and “accession values” mean in this context.

L98 – "rang" -> "range"

L179 – "Shannon-weaver" -> "Shannon-Weaver"

L419:L420 – What is the meaning of “select accessions with minimal number” in this context?

L425 - “It’s” -> “It is”. This is usually not a rule, but contractions should be avoided in research papers. It makes the text too informal.

L432 – How the strategies for creating a core collection based exclusively on the phenotypic traits could affect the maintenance of the adaptative genetic diversity in Miscanthus spp? The use of geographic origin information can help the maintenance of adaptative genetic diversity, but other factors can affect the adaptability of the germplasm. So, what are the best practices to maintain the adaptative genetic diversity in the germplasm?

L476 – What is the meaning of “developing” in this context?

L477 – The word “whose” is used to associate something to someone (person).

Round 2

Reviewer 2 Report

Some of the author's replies satisfied my questions. I consider the manuscript to have merit, and it could be published in Agronomy. However, two main points would still need to be considered before its acceptance:

1) General comment:

The authors committed to making another round of reviews for grammar and spelling. I would like to emphasize that it is a significant step since that would enable a correct judgment of the manuscript's scientific soundness.

2) Discussion:

The discussion about the sampling strategy used to develop a Miscanthus core collection is the key message of the manuscript. The impact of each sampling strategy on the Miscanthus genetic diversity should be explored in the discussion. Therefore, some aspects of the discussion need to be improved.
As mentioned in the first review, adaptative genetic variation is an essential aspect of conserving genetic resources. However, the authors just vaguely explained why their approach would capture adaptative genetic variation in Miscanthus in the discussion. I suggest the authors elaborate the paragraph L449-460 to emphasize the importance of considering adaptative genetic variation in developing a core collection and include why their suggested approach considers adaptative genetic variation and support this discussion by citing other similar works in the literature.
Regarding the development of a core collection and the conservation of genetic resources, since this paper reports an innovative work, the discussion does not need to be always centered in Miscanthus. The authors can discuss the impact of their strategies on the genetic diversity of the whole collections (in general) by comparing and citing core collections developed for other species in the discussion. This would expand the audience's interest in the manuscript since the researchers would also be interested in the sample strategies used in this study, regardless of the species.

Author Response

This manuscript is a resubmission of an earlier submission. The following is a list of the peer review reports and author responses from that submission.

Round 1

Reviewer 1 Report

Dear Authors,

Thank you for this really nice work you exposed in your manuscript. A really good job and I enjoyed reading it. However, I cannot accept your manuscript in present form. Please improve your manuscript according to the following review.

1. Line 44-45: Conservation not conversation

2. Line 89: 37 phenotypic traits were studied includingSPACE14 qualitative traits and 23

3. Table 1: Please correct as follows:

(Does node has hairness?) (Does leaf back has hairness?) (Does sheath has hairness?) (Does sheath mouth has hairness?)(Does internode has waxiness?)(Does node has waxiness?) (Does leaf has waxiness?) (Does sheath has waxiness?)

4. Figures 2,3,4 are too small. Please make them bigger.

(two pictures in one line)

"Fig. 1." and "Fig. 3." - Please change on "Figure 1." and "Figure 3."

5. Line 308: 30%>25%=35%>40%>20%=45%>50%(Table6)

You meant: 30%<25%=35%<40%<20%=45%<50%SPACE(Table6)

The greater and less sings are used incorrectly. Moreover, it seems to me that this is not a good form of the results presentation... It would be better to present them graphically in the form of simple and clear charts.

6. Table 8: I would place Table 8 in the supplementary materials (it's too long).

7. Table 10:  "PD-G20" is bolded. Please change it.

8. Table 10: Please try to present these results graphically in the form of four charts. After that you can put Table 10 to the suplementarny materials.

My congratulations for your very nice work.

Reviewer 2 Report

A review of the literature is carefully and neatly done. I would only mention the oldest literature from the last century - there are many new contemporary ones. Rich and well-described research material. A randomized design of the experiment was answered. Line 122 - no need for a dot in front of the parenthesis. The results are presented clearly descriptively - slightly darkened captions next to the graphs. Table 8 - not too much data for me - maybe some statistical interpretation? Clear and concise conclusions. I recommend it for printing after minor corrections and proofreading by a native speaker.